# Reliability and validity of a newly developed Action Research Arm Test for upper limb function assessment in patients with stroke: A comparison with the conventional version

Daigo Sakamoto[1,2], Toyohiro Hamaguchi[2]*, Yasuhide Nakayama[1], Masahiro Abo[1]*

**1** Department of Rehabilitation Medicine, The Jikei University School of Medicine, Tokyo, Japan,
**2** Department of Rehabilitation, Graduate School of Health Science, Saitama Prefectural University, Saitama, Japan

\* hamaguchi-toyohiro@spu.ac.jp (TH), abo@jikei.ac.jp (MA)

## Abstract

### Background

Following the discontinuation of the conventional Action Research Arm Test (ARAT) import in Japan, a domestically manufactured version with identical assessment protocols but modified equipment was developed. We aimed to evaluate the psycho-metric properties of the newly developed ARAT and compare it with the conventional ARAT in patients with stroke.

### Methods

This single-center cross-sectional study enrolled 64 patients with stroke and hemiparesis who met predefined inclusion criteria. Participants were randomly allocated to a reliability validation (n = 33) or criterion validity validation (n = 31) group. The reliability group underwent duplicate assessments with the new ARAT at 15-min intervals, with video recording for independent inter-rater evaluation. The validity group received the new and conventional ARAT assessments in randomized order. Convergent validity was assessed using Spearman correlation coefficients with Fugl-Meyer Assessment of the Upper Extremity (FMA-UE), Box and Block Test (BBT), grip strength, Motor Activity Log (MAL), and Jikei Assessment Scale for Motor Impairment in Daily Living (JASMID).

### Results

Intra-rater reliability demonstrated excellent agreement (intraclass correlation coefficient [ICC]: 0.997–1.000, 95% confidence interval [CI]: 0.995–1.000) across all subscales and total scores. Inter-rater reliability showed equally excellent agreement (ICC: 0.979–0.999, 95% CI: 0.963–0.999). Bland–Altman analysis revealed limits of agreement within clinically acceptable ranges for all measures. The new ARAT

**Data availability statement:** The data for this study were collected from participants under informed consent that did not include permission for public data sharing or secondary use beyond the scope of this research. Therefore, public sharing of the dataset is not permitted, as even de-identified data must comply with the participants' consent conditions and ethical guidelines. The content, including these requirements, has been approved by the Ethics Committee of the Jikei University School of Medicine (Approval Number 24-222-6988). Anonymized data may be provided by the Clinical Research Support Center, Jikei University School of Medicine, based on reasonable requests for academic research purposes. All requests will be reviewed in accordance with ethical requirements, and data will only be provided to researchers who meet the access criteria. Contact information: Clinical Research Support Center, Jikei University School of Medicine 3-25-8 Nishi-Shimbashi, Minato-ku, Tokyo 105-8461, Japan Tel: +81-3-3433-1111 (Ext. 2187) Fax: +81-3-5400-1388 Email: crb@jikei.ac.jp.

**Funding:** This research was conducted with financial support from the Japan Society for the Promotion of Science (JSPS) Grants-in-Aid for Scientific Research (KAKENHI) (Grant number: JP 24K14384). The funders had no role in study design, data collection and analysis, decision to publish, or preparation of the manuscript.

**Competing interests:** The authors have declared that no competing interests exist.

**Abbreviations:** ADL, Activities of Daily Living; AOU, Amount of Use; ARAT, Action Research Arm Test; BBT, Box and Block Test; FMA-UE, Fugl-Meyer Assessment of the Upper Extremity; ICC, Intraclass Correlation Coefficient; JASMID, Jikei Assessment Scale for Motor Impairment in Daily Living; MDC, Minimal Detectable Change; QOM, Quality of Movement; LOA, Limits of Agreement; MAL, Motor Activity Log; SEM, Standard Error of Measurement

demonstrated very strong convergent validity with FMA-UE ($\rho = 0.934$, $p < 0.001$) and BBT ($\rho = 0.917$, $p < 0.001$), and moderate-to-strong correlations with grip strength ($\rho = 0.683$, $p < 0.001$), MAL subscales ($\rho = 0.610$–$0.666$, $p < 0.001$), and JASMID subscales ($\rho = 0.806$–$0.808$, $p < 0.001$).

## Conclusions

The new ARAT demonstrates measurement properties equivalent to the conventional version with excellent reliability and strong criterion-related validity. Its robust convergent validity with established upper limb assessments supports its clinical utility for comprehensive stroke rehabilitation evaluation.

## 1. Introduction

Stroke remains a leading cause of long-term disability worldwide, with upper limb hemiparesis affecting approximately 80% of stroke survivors and significantly compromising their independence in activities of daily living (ADL) and overall quality of life [1, 2]. The restoration of upper limb function represents a critical rehabilitation goal, as even modest improvements can substantially enhance functional independence and reduce caregiver burden [3]. Effective rehabilitation interventions require accurate assessment tools to guide treatment selection, monitor progress, and evaluate therapeutic efficacy across the continuum of stroke recovery.

Recent advances in neurorehabilitation have introduced promising evidence-based interventions that harness neuroplastic mechanisms to promote upper limb recovery. Noninvasive brain stimulation techniques, including repetitive transcranial magnetic stimulation and transcranial direct current stimulation, have demonstrated significant therapeutic potential when combined with conventional rehabilitation approaches [4, 5]. Similarly, brain–machine interface technologies and robot-assisted therapies have shown efficacy in facilitating motor relearning and functional recovery [6, 7]. However, the successful implementation of these interventions depends critically on the availability of reliable and valid assessment tools that can accurately quantify motor impairment severity, guide treatment intensity, and detect clinically meaningful changes over time.

The Action Research Arm Test (ARAT) has emerged as the gold standard assessment tool for upper limb function evaluation in stroke rehabilitation research and clinical practice [8, 9]. Developed by Lyle in 1981 and extensively validated across diverse populations, the ARAT demonstrates exceptional psychometric properties, with excellent inter-rater reliability (intraclass correlation coefficient [ICC] > 0.95), test–retest reliability (ICC > 0.96), and strong construct validity [10, 11]. Recent international studies have further confirmed its robust measurement properties across different cultural contexts, with successful validations in Spanish, Italian, and Chinese populations [12–14]. The assessment comprises 19 hierarchically ordered tasks across four functional domains: Grasp (6 items), Grip (4 items), Pinch (6 items), and Gross movement (3 items), each scored on a 4-point ordinal scale (0–3), yielding a maximum total score of 57 points [8].

The clinical utility of the ARAT extends beyond its psychometric excellence. Its hierarchical task structure enables efficient administration through standardized decision rules, reducing assessment burden while maintaining measurement precision [15]. The test strongly correlates with real-world upper limb usage patterns as measured by the Motor Activity Log (MAL; $r > 0.70$), demonstrating its ecological validity and clinical relevance [16, 17]. In addition, established cutoff scores enable meaningful interpretation of functional recovery trajectories, with scores of 0–10, 11–21, 22–42, 43–54, and 55–57 points corresponding to no, poor, limited, notable, and full recovery capacity, respectively [18].

In Japan, the ARAT has been widely adopted as the standard upper limb function assessment tool in clinical practice and stroke rehabilitation research. However, the recent discontinuation of the conventional ARAT import has created a significant clinical and research challenge. Inter Reha Co. (Tokyo, Japan) developed a domestically manufactured version to address this gap, maintaining identical assessment procedures and scoring criteria while incorporating some modifications in equipment specifications, including variations in the size, weight, and materials of test objects.

Although these modifications appear minor, evidence from standardized assessment research indicates that even subtle changes in equipment characteristics can influence task difficulty, movement strategies, and measurement precision [19, 20]. Such variations may affect the hierarchical ordering of task difficulty, potentially altering score interpretation and clinical decision-making. Moreover, the introduction of modified equipment without rigorous psychometric validation could compromise the continuity of research data and clinical practice standards that have been established with the conventional ARAT over decades of use.

The transition to a new assessment version requires comprehensive psychometric validation to ensure measurement equivalence and to maintain the scientific rigor established with the original instrument. International guidelines for health measurement tools emphasize the importance of establishing reliability, validity, and measurement agreement when implementing modified versions of established assessments [21, 22]. Recent technological advances have highlighted the potential for remote administration and shortened assessment protocols, making comprehensive validation particularly timely [23, 24].

Therefore, the purpose of this study was to establish the psychometric properties of the newly developed ARAT through a comprehensive evaluation of its intra-rater reliability, inter-rater reliability, criterion-related validity against the conventional ARAT, and convergent validity with established clinical measures in patients with stroke and hemiparesis. This validation will ensure the seamless transition from the conventional to the new ARAT while maintaining the highest standards of measurement precision and clinical utility that have made the ARAT an indispensable tool in stroke rehabilitation worldwide.

## 2. Materials and methods

### 2.1. Study design

This single-center, cross-sectional study evaluated the reliability and validity of a newly developed ARAT compared with the conventional version in patients with stroke and hemiparesis. The study protocol was developed a priori and is available as Supporting Information in both English (S1 Protocol) and Japanese (S2 Protocol). This study is reported in accordance with the CONSORT 2025 statement (S3 Checklist).

### 2.2. Ethical considerations

This study was conducted in accordance with the Declaration of Helsinki and approved by the Ethics Committee of the Jikei University School of Medicine (approval number: 24-222-6988). All participants provided written informed consent prior to participation. The study was registered with the University Hospital Medical Information Network Clinical Trials Registry (ID: UMIN000056693).

## 2.3. Participants

Participants were recruited from patients with stroke and hemiparesis who received occupational therapy at Jikei University Hospital between December 1, 2024, and May 31, 2025. Inclusion criteria were: (1) age ≥ 18 years; (2) ability to maintain an independent sitting position; and (3) diagnosis of stroke with resulting hemiparesis.

Exclusion criteria included: (1) disturbance of consciousness; (2) cognitive impairment (Mini-Mental State Examination score ≤ 25) or a diagnosis of post-stroke cognitive impairment affecting the ability to understand instructions and perform tasks; (3) recurrent stroke; (4) visual field impairment; (5) bilateral upper limb motor paralysis; (6) central nervous system or orthopedic diseases other than stroke; (7) pain in the upper limb or finger joints during movement; (8) marked limitation of upper limb range of motion; (9) amputation of upper limb, hand, or fingers; and (10) missing data.

## 2.4. Sample size calculation

the minimum sample size for reliability testing was calculated based on previous studies [25, 26], assuming ICC = 0.8, minimum acceptable ICC = 0.6, α = 0.05, power (1 − β) = 0.8, with two raters and two measurements per participant. The calculation yielded a requirement of 28–30 participants per group. For criterion-related validity, using G*Power 3.1 (Heinrich Heine University Düsseldorf, Düsseldorf, Germany) [27] with an expected correlation coefficient r = 0.75, α = 0.05, and power = 0.95, the minimum required sample was 16 participants. Accounting for a potential 10% data loss, we targeted 30 participants per group.

## 2.5. Randomization and blinding

Eligible participants were randomly allocated to either the reliability testing group (intra-rater and inter-rater reliability) or the validity testing group (criterion-related validity) using a random-number table. Group allocation and assessment-order randomization were performed by a research coordinator who was not involved in data collection, using computer-generated random-number sequences. For inter-rater reliability testing, the second examiner was blinded to the first examiner's scores and participants' clinical characteristics throughout the video-based scoring process.

## 2.6. Setting

The study was conducted at Jikei University Hospital, a tertiary care university hospital in Tokyo, Japan, that provides advanced medical care. Data collection, including clinical evaluations and motor function assessments, was conducted from December 1, 2024, to May 31, 2025.

## 2.7. Instruments

### 2.7.1. Action Research Arm Test.
The ARAT is an upper limb function assessment tool for patients with stroke [8], consisting of four subscales—Grasp, Grip, Pinch, and Gross movement—with a total of 19 tasks. Each task is scored on a 4-point ordinal scale, ranging from 0 (unable to perform) to 3 (normal performance), with a maximum total score of 57 points.

The assessment follows a hierarchical structure: if the first task of a subscale receives full marks, the entire subscale is scored at the maximum, and the remaining tasks are omitted. Conversely, if the second task scores 0 points in the Grasp, Grip, and Pinch subscales, or if the first task scores 0 points in the Gross movement, the subsequent tasks are omitted, and the entire subscale is scored as 0 points.

The new and conventional ARAT versions use identical assessment protocols and scoring methods, differing only in equipment specifications and materials. The conventional ARAT was manufactured by Reha-Stim MedTec AG (Schlieren, Switzerland), whereas the new ARAT was developed and manufactured by Inter Reha Co. (S4 Appendix). The new ARAT maintained identical assessment protocols but incorporated modifications in several test objects. Key differences include

variations in material composition (wood vs. composite materials), object dimensions (±2–5 mm in some items), and surface textures. Complete specifications are provided in S5 and S6 Tables.

**2.7.2. Experimental procedures. Reliability testing group.** For intra-rater reliability testing, a single examiner administered the new ARAT twice, with a 15-min interval between administrations. No therapeutic intervention was provided between measurements. During the first assessment, the examiner video-recorded the participant's performance for all 19 tasks, although scoring followed standard ARAT procedures, including task omission rules.

For inter-rater reliability testing, a second examiner scored the video recordings without knowledge of the first examiner's scores. Examiner assignments were randomized using a random number table. The second examiner applied the standard ARAT scoring procedures, including task omission rules based on performance levels.

For inter-rater reliability assessment, video recordings were obtained using a standardized setup, with cameras positioned to ensure clear visualization of all hand and arm movements during task performance. Recording began before task instruction delivery and continued until task completion. Camera angles were optimized to capture sagittal and frontal plane movements while ensuring clear visibility of object manipulation.

**Validity testing group.** For criterion-related validity testing, a single examiner administered the new and conventional ARAT versions, with a 15-min interval between administrations. No therapeutic intervention was provided between assessments. The order of administrations was randomized using a random number table.

**Clinical evaluations.** The primary measures for reliability and validity testing were the new ARAT total and subscale scores. As secondary outcome measures, motor impairment was assessed using the Fugl-Meyer Assessment of the Upper Extremity (FMA-UE) [28], which evaluates upper limb motor function on a 3-point ordinal scale with a maximum score of 66 points. Manual dexterity of the paretic hand was evaluated using the Box and Block Test (BBT) [29], which measures the number of blocks moved within 1 min. Grip strength on the paretic side was measured using a Jamar-type dynamometer, with three measurements taken and the average calculated.

Sensory function was assessed by evaluating superficial sensation in the paretic arm and fingers using a 3-point ordinal scale, where 0 indicates absent sensation, 1 indicates impaired sensation, and 2 indicates normal sensation. Self-reported measures of upper limb use in ADL were collected using two instruments. The MAL [16] was used to assess Amount of Use (AOU) and Quality of Movement (QOM) on 5-point scales, with each domain scored to a maximum of 100 points. Additionally, the Jikei Assessment Scale for Motor Impairment in Daily Living (JASMID) [30], which was developed based on the MAL and specifically adapted for the Japanese lifestyle, was used. JASMID similarly assesses the quantity and quality of paretic upper limb use, with each domain scored to a maximum of 100 points.

**Participant characteristics.** Demographic and medical information collected included sex, age, height, weight, body mass index, pre-stroke dominant hand, stroke type (ischemic or hemorrhagic), stroke location, time since stroke onset, and paretic side.

Investigators

Clinical evaluations were conducted by seven occupational therapists affiliated with Jikei University Hospital, who had a mean clinical experience of 10.0 ± 6.2 years. All examiners had completed ARAT training workshops conducted by the Japanese Stimulation Therapy Society and possessed sufficient knowledge of and experience with ARAT and other clinical assessments. Prior to study initiation, all examiners received ARAT evaluation manuals and participated in standardization sessions to ensure consistent assessment procedures and scoring methods.

**2.7.3. Statistical analysis. Descriptive statistics.** Participant characteristics and clinical evaluation data were analyzed using descriptive statistics. Data normality was assessed using the Shapiro–Wilk test. Normally distributed data were reported as means ± standard deviations, whereas non-normally distributed data were reported as medians with interquartile ranges. For descriptive purposes, baseline characteristics between the reliability verification group and the validity verification group were confirmed using Student's t-test or the Mann–Whitney U test, as appropriate.

**Reliability analysis.** Intra-rater reliability was assessed using the ICC model (2,1) for two measurements by the same examiner. Inter-rater reliability was assessed using the ICC (2,1) for measurements by different examiners [31]. ICC values ≥ 0.75 were interpreted as excellent agreement. Measurement error was quantified using the Standard Error of Measurement (SEM):

$$SEM = \text{standard deviation} \times \sqrt{(1-ICC)}$$

Values within 10% of the score range were considered acceptable. The Minimal Detectable Change ($MDC_{95}$) was calculated as:

$$MDC_{95} = SEM \times 1.96 \times \sqrt{2}$$

Results were rounded to the nearest integer for clinical interpretation, with values <1 point reported as 1 point (minimum clinical unit).

**Agreement analysis.** Item-level agreement was assessed using weighted kappa coefficients with quadratic weighting for intra-rater and inter-rater comparisons of the new ARAT. The weighted kappa coefficients were interpreted as follows [32]: <0.00 = poor, 0.00–0.20 = slight, 0.21–0.40 = fair, 0.41–0.60 = moderate, 0.61–0.80 = substantial, and 0.81–1.00 = almost perfect. In addition, the Overall Agreement, defined as the percentage of evaluators who selected the same answer, was calculated.

**Validity analysis.** Criterion-related validity between the new and conventional ARAT was assessed using Bland–Altman analysis [33], which calculated bias and 95% limits of agreement (LOA). Clinical acceptability was defined as an LOA within the absolute value of $MDC_{95}$. Bland–Altman plots were generated to visually assess systematic bias, proportional bias, outliers, and agreement limits.

Convergent validity was assessed by calculating correlation coefficients between the new ARAT total scores and the FMA-UE total scores, paretic-side BBT and grip strength, MAL, and JASMID scores. Pearson's correlation coefficient was used for normally distributed data, and Spearman's rank correlation coefficient was used for non-normally distributed data. For convergent validity analysis, the new ARAT total scores were obtained from both study groups: first-trial measurements from the reliability testing group (n = 33) and single measurements from the validity testing group (n = 31), resulting in a combined sample of 64 participants for correlation analysis with other clinical measures. Furthermore, correlation analyses were conducted separately for the reliability testing group and the validity testing group.

Correlation strength was interpreted as follows [34]: <0.4 = weak, 0.4–0.74 = moderate, 0.75–0.9 = strong, and ≥0.9 = very strong. All analyses were performed using Jamovi version 2.6.44 (https://www.jamovi.org). Statistical significance was set at $p < 0.05$.

## 3. Results

A total of 65 patients who met the eligibility criteria were approached for participation. After providing informed consent and undergoing randomization, 33 participants were allocated to the reliability testing group and 31 to the validity testing group, resulting in a final analyzed sample of 64 participants with no missing data (Fig 1).

Participant clinical characteristics are presented in Table 1. The two groups showed comparable baseline demographics and stroke characteristics. Complete data were available for all participants across all measured variables. In the reliability testing group, stroke locations for cerebral infarction included: middle cerebral artery territory (n = 8), brainstem (n = 3), internal capsule (n = 3), corona radiata (n = 2), and thalamus (n = 1). Intracerebral hemorrhage locations comprised: putamen (n = 12), subcortical (n = 2), and thalamus (n = 2). The validity testing group showed similar distribution patterns. In the validity testing group, stroke locations for cerebral infarction included: middle cerebral artery region infarction (n = 7), corona radiata (n = 3), thalamus (n = 2), and brainstem (n = 1). Intracerebral hemorrhage locations comprised: putamen (n = 13), subcortical (n = 3), and thalamus (n = 2).

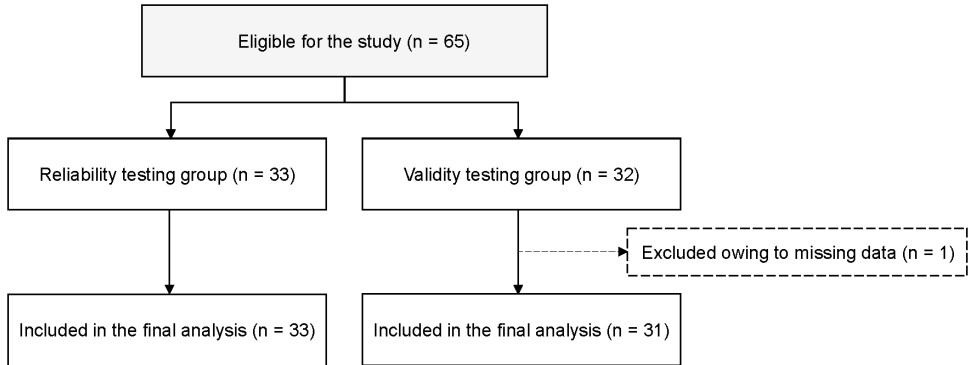

**Fig 1. Patient selection process.** Diagram showing the selection process of participants through the study. After allocation, 33 and 31 participants were analyzed for reliability and validity testing, respectively.

Raw ARAT scores demonstrated appropriate distribution across the measurement range, with median total scores of 36 points in the reliability group and 7–8 points in the validity group, indicating that participants across the spectrum of upper limb impairment severity were included (Table 2).

Reliability analysis results are presented in Table 3. Intra-rater and inter-rater reliability demonstrated excellent agreement, with ICC values ranging from 0.979 to 1.000 (95% confidence interval [CI]: 0.963–1.000) across all subscales and total scores. For intra-rater reliability, the total score ICC approached perfect agreement (ICC = 1.000, rounded from 0.9995). All SEM values were within the acceptable threshold of less than 10% of the score range, with $MDC_{95}$ values ranging from 1 to 4 points across subscales.

Item-level agreement analysis revealed weighted kappa coefficients of ≥0.90 for all 19 ARAT items in intra-rater and inter-rater comparisons, indicating almost perfect agreement (Tables 4 and 5). Perfect agreement (κ = 1.00) was achieved for six items in intra-rater analysis and two items in inter-rater analysis. Overall percentage agreement typically exceeded 90%, with three items showing lower agreement rates: Grasp Block (7.5 cm³) at 57.6% (intra-rater), Pinch Marble with index finger and thumb at 54.6% (intra-rater), and Grip Washer over bolt at 42.4% (inter-rater). Despite a lower percentage agreement, these items maintained excellent weighted kappa values (≥ 0.95), indicating that disagreements were limited to adjacent score categories.

Criterion-related validity between the new and conventional ARAT was assessed using Bland–Altman analysis (Table 6). Mean bias values were generally minimal across subscales and total scores (range: –0.091 to 0.303 points). Notably, the Grasp subscale in the inter-rater comparison showed a relatively higher bias (0.303) compared with other subscales (range: –0.091 to 0.030), although this value remained within clinically acceptable limits. Despite this finding, the 95% limits of agreement for all measures, including Grasp, fell within the clinically acceptable range defined by MDC values, confirming measurement equivalence between the two versions. Specifically, LOA ranges (rounded for clinical interpretation) were as follows: Grasp (–1–1), Grip (–1–1 for intra-rater, –1–2 for inter-rater), Pinch (–1–1), Gross movement (–1–1), and Total score (–1–1 for intra-rater, –1–2 for inter-rater).

Visual inspection of the Bland–Altman plots (Figs 2 and 3) revealed no systematic bias or proportional bias patterns. Data points were randomly distributed around the mean difference line, with most observations falling within the LOA boundaries. A few outliers were observed, but they did not indicate systematic measurement issues.

Convergent validity was examined through correlation analysis using Spearman's rank correlation coefficient. The new ARAT total score demonstrated very strong positive correlations with established motor assessments: FMA-UE total score (ρ = 0.934, p < 0.001) and paretic-side BBT (ρ = 0.917, p < 0.001) (Fig 4). A moderate positive correlation

**Table 1. Clinical characteristics of the analyzed patients.**

| Characteristics | | | Reliability testing group (n = 33) | Validity testing group (n = 31) | p |
|---|---|---|---|---|---|
| Age (years) | | | 54 ± 14 | 58 ± 14 | 0.200 |
| Sex | Female: Male | | 13:20 | 13:18 | 0.839 |
| Height (cm) | | | 162 (157, 170) | 165 (158, 171) | 0.177 |
| Weight (kg) | | | 65 ± 14 | 63 ± 12 | 0.550 |
| Body mass index (kg/m$^2$) | | | 23 (20, 27) | 23 (20, 25) | 0.118 |
| Paretic hand | Left: Right | | 11:22 | 16:15 | 0.143 |
| Dominant hand | Left: Right | | 0:33 | 0:31 | – |
| Laterality in the paretic and dominant hand | Ipsilateral side | | 22 | 15 | 0.143 |
| | Contralateral side | | 11 | 16 | |
| Diagnosis | CI | | 17 | 13 | 0.451 |
| | ICH | | 16 | 18 | |
| Post-onset period | Early (days) | | 6; 15 (13, 17) | 1; 55 | 0.342 |
| | Chronic (months) | | 27; 101 (60, 140) | 30; 72 (43, 150) | |
| FMA-UE | Total | | 56 (35, 61) | 35 (25, 53) | 0.022 |
| BBT | Paretic side | | 13 (0, 36) | 0 (0, 17) | 0.028 |
| Grip strength (kg) | Paretic side | | 6 (2, 11) | 2 (0, 7) | 0.050 |
| Sense of touch | Finger | Normal | 16 | 16 | 0.806 |
| | | Decline | 17 | 15 | |
| | Arm | Normal | 16 | 17 | 0.618 |
| | | Decline | 17 | 14 | |
| JASMID | Quantity | | 53 (23, 77) | 29 (21, 52) | 0.153 |
| | Quality | | 37 (22, 58) | 29 (21, 50) | 0.353 |
| MAL | Amount of Use | | 2.1 (1.1, 3.4) | 1.3 (0.7, 3.0) | 0.172 |
| | Quality of Movement | | 2.3 (1.0, 3.2) | 1.5 (0.4, 2.7) | 0.130 |

Values are presented as n, mean ± standard deviation, or median (25th, 75th percentile). The ARAT scores were the first measurement results of the reliability group (n = 33) and the measurement results of the new ARAT in the validity group (n = 31). The post-onset period is classified into early phase (<6 months after onset) and chronic phase (more than 6 months after onset). Group comparisons were performed using the unpaired t-test or the Mann–Whitney U test, as appropriate. –: Not applicable; all participants were right-handed, and therefore statistical comparison was not possible.

ARAT, Action Research Arm Test; BBT, Box and Block Test; CI, cerebral infarction; FMA-UE, Fugl-Meyer Assessment of the Upper Extremity; ICH, intra-cranial hemorrhage; JASMID, Jikei Assessment Scale for Motor Impairment in Daily Living; MAL, Motor Activity Log.

**Table 2. Raw data for Action Research Arm Test total scores and subscale scores.**

| Scale | Reliability testing group (n = 33) | | | Validity testing group (n = 31) | |
|---|---|---|---|---|---|
| | New ARAT | | | New ARAT | Conventional ARAT |
| | 1st Trial | 2nd Trial | 2nd Rater | | |
| Grasp | 13 (1, 18) | 14 (1, 17) | 11 (1, 17) | 3 (0, 12) | 3 (0, 12) |
| Grip | 8 (0, 12) | 8 (1, 12) | 8 (0, 12) | 1 (0, 8) | 1 (0, 8) |
| Pinch | 7 (0, 17) | 9 (0, 17) | 7 (0, 17) | 0 (0, 9) | 0 (0, 10) |
| Gross movement | 6 (4, 9) | 6 (4, 9) | 6 (4, 9) | 4 (3, 7) | 4 (3, 7) |
| Total | 36 (7, 52) | 36 (7, 52) | 37 (6, 52) | 7 (3, 36) | 8 (3, 37) |

Values are presented as median (25th, 75th percentile).

ARAT, Action Research Arm Test.

**Table 3. Reliability of the total score and the subscale scores of the Action Research Arm Test.**

| Scale | Intra-rater reliability | | | Inter-rater reliability | | |
|---|---|---|---|---|---|---|
| | ICC (95% CI lower, upper) | SEM | MDC$_{95}$ | ICC (95% CI lower, upper) | SEM | MDC$_{95}$ |
| Grasp (out of 18) | 0.997 (0.995, 0.998) | 0.426 | 1 | 0.999 (0.998, 0.999) | 0.912 | 3 |
| Grip (out of 12) | 0.998 (0.997, 0.999) | 0.225 | 1 | 0.988 (0.977, 0.994) | 0.555 | 2 |
| Pinch (out of 18) | 0.998 (0.997, 0.999) | 0.353 | 1 | 0.999 (0.998 0.999) | 1.365 | 4 |
| Gross movement (out of 9) | 0.997 (0.995, 0.999) | 0.132 | 1 | 0.979 (0.963, 0.989) | 0.424 | 1 |
| Total (out of 57) | 1.000 (0.999, 1.000) | 0.486 | 1 | 0.999 (0.998, 0.999) | 0.685 | 2 |

MDC$_{95}$ computed as $1.96 \times \sqrt{2} \times$ SEM; SEM = standard deviation $\times \sqrt{(1 - \text{ICC})}$. For the total score, although ICC is reported as 1.000 in the table (rounded), full-precision ICC (0.9995) was used for SEM/MDC$_{95}$ calculations.

CI, Confidence Interval; MDC, Minimal Detectable Change; ICC, intraclass correlation coefficient; SEM, Standard Error of Measurement.

**Table 4. Intra-rater reliability of ARAT subscales assessed using weighted kappa coefficients.**

| Subtest | Test | Item | Weighted kappa | 95% CI (lower, upper) | p | Overall Agreement (%) |
|---|---|---|---|---|---|---|
| Grasp | 1 | Block (10 cm³) | 0.97 | 0.63, 1.31 | <0.001 | 90.9 |
| | 2 | Block (2.5 cm³) | 0.95 | 0.61, 1.29 | <0.001 | 93.9 |
| | 3 | Block (5 cm³) | 0.99 | 0.65, 1.33 | <0.001 | 97.0 |
| | 4 | Block (7.5 cm³) | 0.99 | 0.65, 1.33 | <0.001 | 57.6 |
| | 5 | Cricket ball (diameter, 7.5 cm) | 1.00 | 0.66, 1.34 | <0.001 | 100.0 |
| | 6 | Sharpening stone (10 × 2.5 × 1 cm) | 0.99 | 0.65, 1.33 | <0.001 | 97.0 |
| Grip | 7 | Pour water from glass to glass | 1.00 | 0.66, 1.34 | <0.001 | 100.0 |
| | 8 | Tube (diameter, 2.25 cm) | 0.98 | 0.64, 1.32 | <0.001 | 93.9 |
| | 9 | Tube (diameter, 1 cm) | 0.99 | 0.65, 1.33 | <0.001 | 97.0 |
| | 10 | Washer over bolt | 0.98 | 0.64, 1.32 | <0.001 | 97.0 |
| Pinch | 11 | Ball bearing (diameter, 6 mm), ring finger and thumb | 0.96 | 0.62, 1.30 | <0.001 | 87.9 |
| | 12 | Marble (diameter, 1.5 cm), index finger and thumb | 0.99 | 0.65, 1.33 | <0.001 | 54.6 |
| | 13 | Ball bearing (diameter, 6 mm), middle finger and thumb | 1.00 | 0.66, 1.34 | <0.001 | 100.0 |
| | 14 | Ball bearing (diameter, 6 mm), index finger and thumb | 0.99 | 0.65, 1.33 | <0.001 | 97.0 |
| | 15 | Marble (diameter, 1.5 cm), ring finger and thumb | 0.98 | 0.64, 1.33 | <0.001 | 93.9 |
| | 16 | Marble (diameter, 1.5 cm), middle finger and thumb | 1.00 | 0.66, 1.34 | <0.001 | 100.0 |
| Gross movement | 17 | Place hand behind head | 1.00 | 0.66, 1.34 | <0.001 | 100.0 |
| | 18 | Place hand on top of head | 1.00 | 0.66, 1.34 | <0.001 | 100.0 |
| | 19 | Hand to mouth | 0.98 | 0.64, 1.32 | <0.001 | 97.0 |

The weighted kappa coefficients were interpreted as follows: < 0.00 = poor, 0.00–0.20 = slight, 0.21–0.40 = fair, 0.41–0.60 = moderate, 0.61–0.80 = substantial, and 0.81–1.00 = almost perfect.

CI, Confidence Interval

was observed with paretic-side grip strength ($\rho = 0.683$, $p < 0.001$). Self-reported activity measures showed significant correlations: moderate correlations with MAL AOU ($\rho = 0.610$, $p < 0.001$) and QOM ($\rho = 0.666$, $p < 0.001$), and strong correlations with JASMID Quantity ($\rho = 0.806$, $p < 0.001$) and Quality ($\rho = 0.808$, $p < 0.001$) (Fig 5). All correlations were statistically significance, supporting the construct validity of the new ARAT. The results of the group-specific correlation analyses are presented in S7 Table.

**Table 5. Inter-rater reliability of ARAT subscales assessed using weighted kappa coefficients.**

| Subtest | Test | Item | Weighted kappa | 95% CI (lower, upper) | p | Overall Agreement (%) |
|---|---|---|---|---|---|---|
| Grasp | 1 | Block (10 cm³) | 0.97 | 0.63, 1.31 | <0.001 | 90.9 |
| | 2 | Block (2.5 cm³) | 1.00 | 0.66, 1.34 | <0.001 | 100.0 |
| | 3 | Block (5 cm³) | 1.00 | 0.66, 1.34 | <0.001 | 100.0 |
| | 4 | Block (7.5 cm³) | 0.97 | 0.63, 1,31 | <0.001 | 93.9 |
| | 5 | Cricket ball (diameter, 7.5 cm) | 0.99 | 0.65, 1.33 | <0.001 | 97.0 |
| | 6 | Sharpening stone (10 × 2.5 × 1 cm) | 0.98 | 0.63, 1.32 | <0.001 | 91.0 |
| Grip | 7 | Pour water from glass to glass | 0.94 | 0.60, 1.28 | <0.001 | 91.0 |
| | 8 | Tube (diameter, 2.25 cm) | 0.95 | 0.61, 1.29 | <0.001 | 91.0 |
| | 9 | Tube (diameter, 1 cm) | 0.98 | 0.64, 1.32 | <0.001 | 93.9 |
| | 10 | Washer over bolt | 0.95 | 0.61, 1.29 | <0.001 | 42.4 |
| Pinch | 11 | Ball bearing (diameter, 6 mm), ring finger and thumb | 0.99 | 0.65, 1.33 | <0.001 | 97.0 |
| | 12 | Marble (diameter, 1.5 cm), index finger and thumb | 0.94 | 0.60, 1.29 | <0.001 | 90.6 |
| | 13 | Ball bearing (diameter, 6 mm), middle finger and thumb | 0.99 | 0.65, 1.33 | <0.001 | 97.0 |
| | 14 | Ball bearing (diameter, 6 mm), index finger and thumb | 0.99 | 0.65, 1.33 | <0.001 | 97.0 |
| | 15 | Marble (diameter, 1.5 cm), ring finger and thumb | 0.99 | 0.65, 1,33 | <0.001 | 97.0 |
| | 16 | Marble (diameter, 1.5 cm), middle finger and thumb | 0.98 | 0.64, 1.33 | <0.001 | 93.9 |
| Gross movement | 17 | Place hand behind head | 0.94 | 0.60, 1.28 | <0.001 | 90.9 |
| | 18 | Place hand on top of head | 0.94 | 0.60, 1.28 | <0.001 | 90.9 |
| | 19 | Hand to mouth | 0.91 | 0.57, 1.25 | <0.001 | 87.9 |

The weighted kappa coefficients were interpreted as follows: <0.00 = poor, 0.00–0.20 = slight, 0.21–0.40 = fair, 0.41–0.60 = moderate, 0.61–0.80 = substantial, and 0.81–1.00 = almost perfect.

CI, Confidence Interval

**Table 6. Bland–Altman analysis of total and subscale scores between the new and original Action Research Arm Test.**

| Scale | Intra-rater | | | Inter-rater | | |
|---|---|---|---|---|---|---|
| | Bias | SD | 95% LOA (lower, upper) | Bias | SD | 95% LOA (lower, upper) |
| Grasp | 0.030 | 0.585 | −1, 1 | 0.030 | 0.394 | −1, 1 |
| Grip | −0.030 | 0.305 | −1, 1 | 0.303 | 0.728 | −1, 2 |
| Pinch | 0.000 | 0.500 | −1, 1 | 0.030 | 0.394 | −1, 1 |
| Gross movement | −0.030 | 0.174 | −1, 1 | 0.000 | 0.500 | −1, 1 |
| Total | −0.091 | 0.678 | −1, 1 | 0.303 | 0.918 | −1, 2 |

LOA, limits of agreement; SD, standard deviation.

The mean difference represents the average of the differences between the new and conventional Action Research Arm Test scores. LOA was calculated as the mean difference ± 1.96 × standard deviation of differences.

## 4. Discussion

This study established the psychometric properties of the newly developed ARAT through a comprehensive evaluation of its reliability and validity in patients with stroke and hemiparesis. The findings demonstrate that the new ARAT maintains measurement characteristics equivalent to those of the conventional version, supporting its clinical utility and research applications.

The excellent intra-rater (ICC: 0.997–1.000) and inter-rater (ICC: 0.979–0.999) reliability values observed in this study are consistent with previous ARAT reliability studies, confirming the reproducibility of the new version [10, 35].

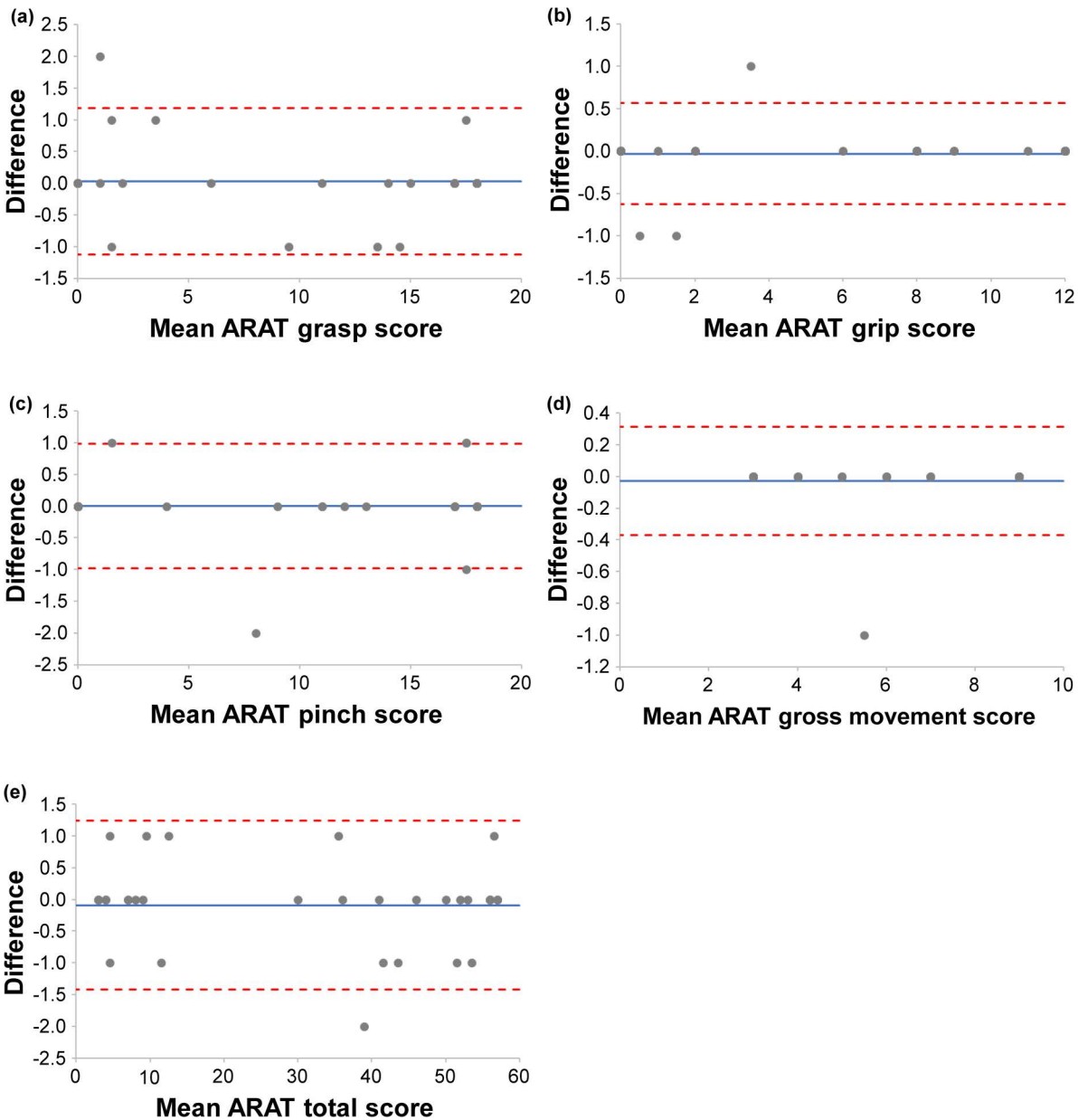

**Fig 2. Bland–Altman plots for intra-rater agreement of the Action Research Arm Test (ARAT) subscales and total score.** The Bland–Altman plots show **(a)** ARAT grasp score, **(b)** ARAT grip score, **(c)** ARAT pinch score, **(d)** ARAT gross movement score, and **(e)** ARAT total score. The solid blue line indicates bias, and the red dashed lines indicate the upper and lower LOA, which were drawn using exact calculated values before rounding. A systematic error is suggested when the bias shifts upward or downward, and a proportional bias is suggested when the difference changes linearly with the mean. Points outside the LOA indicate outliers. Agreement is considered clinically acceptable when most points lie within the LOA, and the LOA width falls within the acceptable range. ARAT, Action Research Arm Test; LOA, Limits of Agreement.

The weighted kappa coefficients (≥0.90) for all items indicated almost perfect agreement at the item level. Although some items showed lower overall agreement percentages, the high weighted kappa values suggest that disagreements were typically limited to within one scoring point, which is clinically acceptable. The items with lower overall agreement (Grasp Block 7.5 cm$^3$, Pinch Marble with the index finger and thumb, and Grip Washer over bolt) should be recognized as

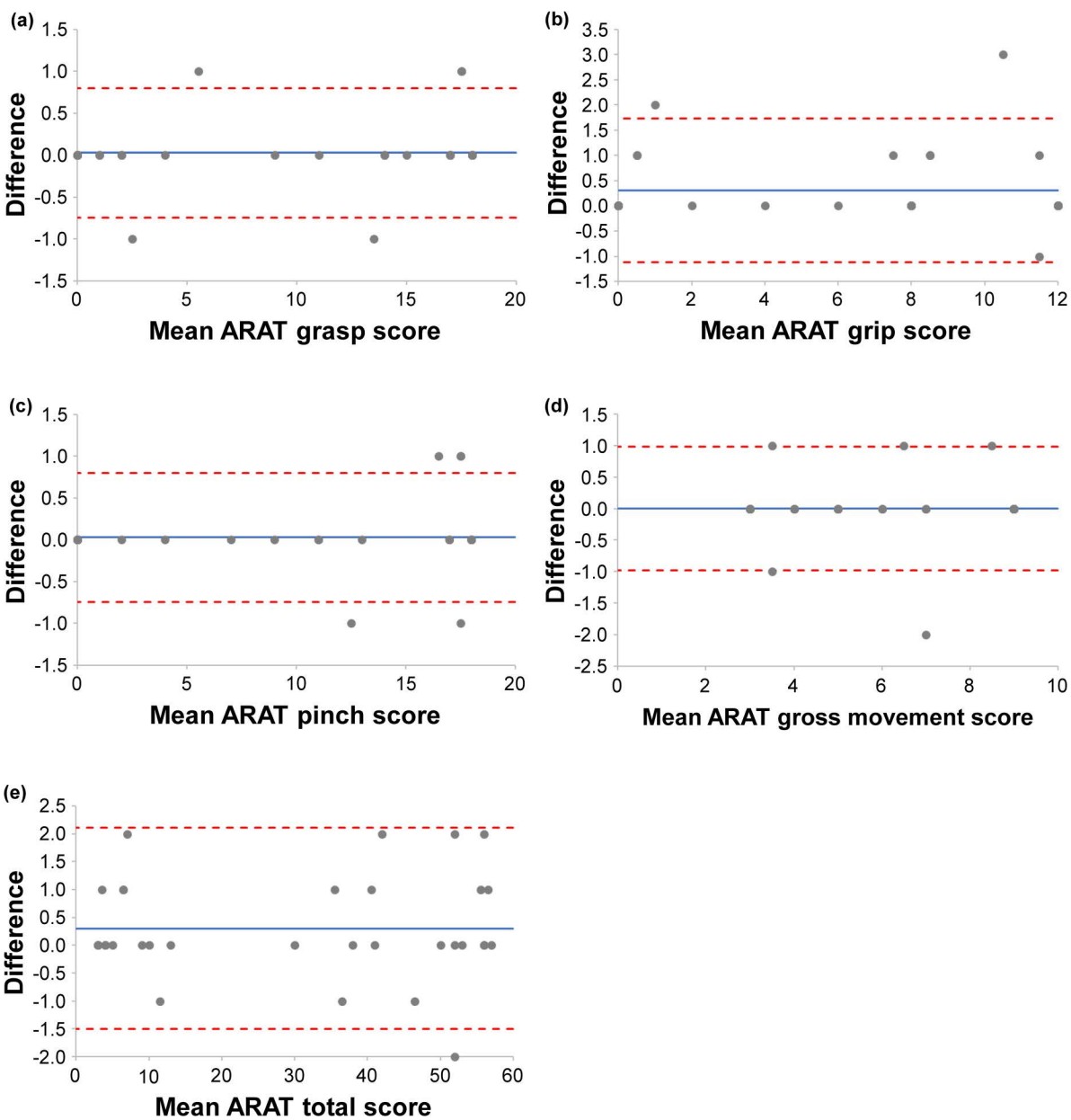

**Fig 3. Bland–Altman plots for inter-rater agreement of the Action Research Arm Test subscales and total score.** The Bland–Altman plots show **(a)** ARAT grasp score, **(b)** ARAT grip score, **(c)** ARAT pinch score, **(d)** ARAT gross movement score, and **(e)** ARAT total score. The solid blue line indicates bias, and the red dashed lines indicate the upper and lower LOA, which were drawn using exact calculated values before rounding. A systematic error is suggested when the bias shifts upward or downward, and a proportional bias is suggested when the difference changes linearly with the mean. Points outside the LOA indicate outliers. Agreement is considered clinically acceptable when most points lie within the LOA, and the LOA width falls within the acceptable range. ARAT, Action Research Arm Test; LOA, Limits of Agreement.

particularly challenging and therefore require careful observation and the application of standardized scoring criteria [36]. However, even in these tasks, the weighted kappa coefficient exceeded 0.9, indicating sufficient reliability, and the differences were limited to within one point in most cases. Overall, the new ARAT produced stable measurement results similar to those of the previous version, confirming reproducibility.

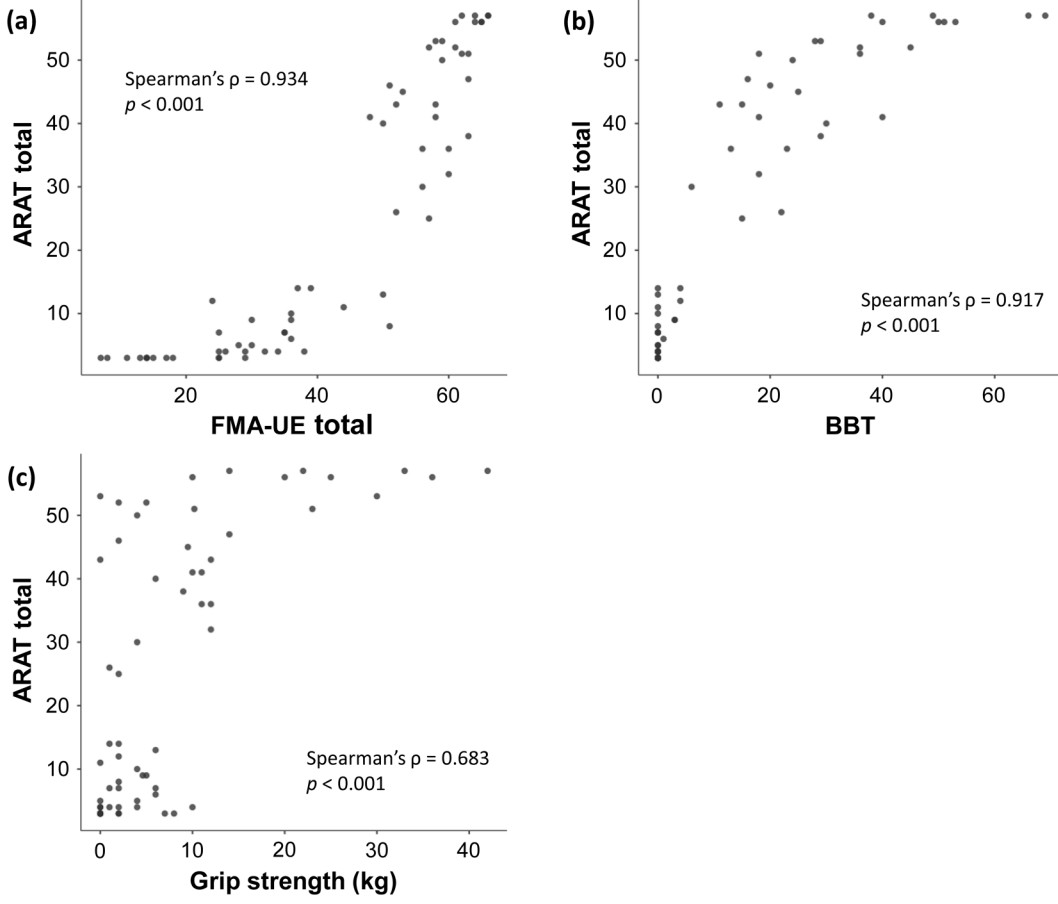

**Fig 4. Correlation between the Action Research Arm Test and upper limb function assessment.** The scatter plot shows the relationship between the ARAT total score and (a) the FMA-UE total score, (b) the BBT on the paretic side, and (c) grip strength on the paretic side. The strength of the correlation was interpreted based on the absolute value of the correlation coefficient as follows: < 0.4 = weak, 0.4–0.74 = moderate, 0.75–0.9 = strong, and ≥ 0.9 = very strong. ARAT, Action Research Arm Test; BBT, Box and Block Test; FMA-UE, Fugl-Meyer Assessment of the Upper Extremity.

Bland–Altman analysis demonstrated excellent agreement between the new and conventional ARAT versions, with minimal systematic bias and limits of agreement falling within the calculated MDC$_{95}$ ranges. This finding indicates that the observed differences between versions are within measurement error and thus clinically insignificant. The small MDC$_{95}$ values (1–4 points) calculated in this study are consistent with the precision expected for ARAT measurements in stroke populations [37, 38]. Although the new ARAT maintained identical assessment protocols and scoring criteria compared with the conventional version, some test objects were modified in size and materials. These changes were implemented to ensure cost-effective manufacturing and long-term stable supply, which are essential for sustained clinical availability in Japan after discontinuing the conventional ARAT import. The Bland–Altman analysis revealed that the Grasp subscale in the inter-rater comparison exhibited a relatively higher bias (0.303) than other subscales. The observed bias in Grasp tasks likely reflects the sensitivity of grasping movements to subtle differences in test object characteristics between the new and conventional ARAT versions. Among the modified objects, grasp-related items showed the greatest variations in dimensions (2–5 mm; see S2 and S3 Tables) and surface textures, which may have influenced tactile feedback, grip stability, and perceived task difficulty. These physical differences could affect patients' motor execution strategies and

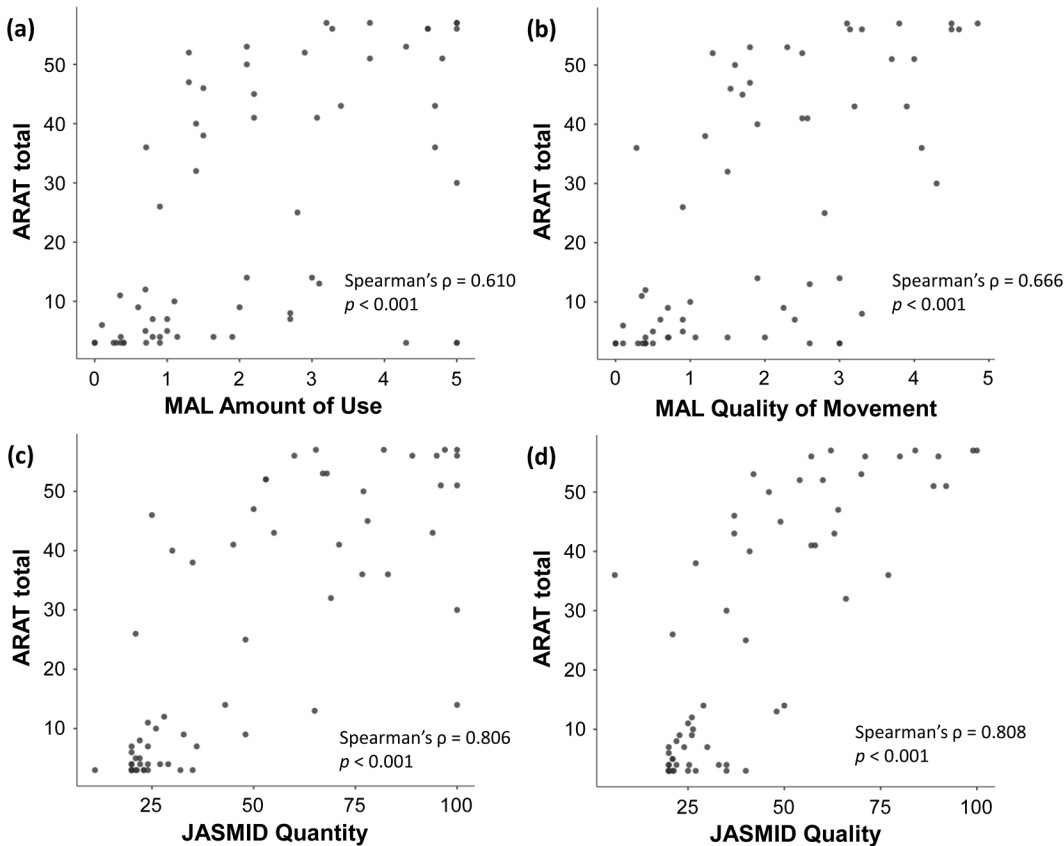

**Fig 5. Correlation between the Action Research Arm Test and self-reported outcomes**. The scatter plot shows the relationship between the ARAT total score and **(a)** MAL Amount of Use, **(b)** MAL Quality of Movement, **(c)** JASMID Quantity, and **(d)** JASMID Quality. The strength of the correlation was interpreted based on the absolute value of the correlation coefficient as follows: < 0.4 = weak, 0.4–0.74 = moderate, 0.75–0.9 = strong, and ≥0.9 = very strong. ARAT, Action Research Arm Test; JASMID, Jikei Assessment Scale for Motor Impairment in Daily Living; MAL, Motor Activity Log.

evaluators' scoring judgments, resulting in the observed systematic difference. However, this bias remained within clinically acceptable limits, as the LOA fell within the $MDC_{95}$ range, indicating preserved measurement equivalence with no meaningful impact on decision-making. Overall, the minor equipment modifications required for domestic manufacturing and stable supply did not substantially affect the measurement properties that have been maintained for decades with the conventional ARAT. The convergent validity findings support the construct validity of the new ARAT. Very strong correlations with FMA-UE ($\rho = 0.934$) and BBT ($\rho = 0.917$) confirm that the new ARAT effectively captures upper limb motor impairment and manual dexterity, consistent with previous validation studies [38, 39]. The moderate correlation with grip strength ($\rho = 0.683$) appropriately reflects that ARAT tasks primarily assess coordination and functional movement patterns rather than maximum force generation [40]. Grip strength is a known predictor of functional prognosis after stroke [41, 42], and previous studies have reported that a combined assessment of ARAT grasp performance and grip strength is useful for predicting upper limb function outcomes [43]. This predictive value is presumed to derive from the complementary information provided by the two measures: whereas the ARAT grasp task ($\rho = 0.683$) evaluates functional object manipulation under standardized conditions, grip strength quantifies maximal voluntary muscle force. The moderate correlation observed in this study indicates that these measures assess related but distinct dimensions of upper limb function, supporting their combined use for comprehensive clinical evaluation and individualized goal setting. The correlations with

self-reported measures varied appropriately: MAL showed moderate correlations ($\rho = 0.610$–0.666), whereas JASMID demonstrated stronger correlations ($\rho = 0.806$–0.808). This difference may reflect the adaptation of JASMID to Japanese lifestyle patterns and its sensitivity to detecting functional changes in the study population [30]. These findings confirm that the new ARAT maintains its established relationship with objective motor assessments and perceived upper limb function in daily activities. Scatter plots presented in Figs 4 and 5 revealed clustering patterns in which patients with similar ARAT scores exhibited substantial variability in other clinical measures, and vice versa. This heterogeneity was particularly evident in the low-score range (ARAT < 10) and the high-score range (ARAT > 50), where patients clustered at discrete ARAT values yet showed wide dispersion in FMA-UE, BBT, and self-reported outcomes. These patterns suggest that ARAT scores reflect multidimensional aspects of upper limb function, and that patients achieving comparable ARAT performance may differ considerably in underlying motor capacity (FMA-UE), manual dexterity (BBT), maximal force generation (grip strength), and perceived functional use (MAL/JASMID).

This study has some limitations. First, it was conducted at a single facility, thus restricting the generalizability of the findings to a wide range of patients with stroke. Many of the participants were in the chronic phase; therefore, the characteristics of patients in the acute and subacute phases were not fully reflected. Immediately after onset, patients may have different upper limb motor strategies compared to those in the chronic phase [44]. Regarding this point, additional verification involving acute-phase patients at multiple facilities is required. Second, all evaluators in this study had sufficient experience in measuring ARAT, suggesting that their measurement techniques contributed to high reliability. Therefore, careful consideration is needed to determine whether evaluators with less experience in clinical settings can achieve similar reliability. Third, this study was conducted using a cross-sectional design, and reactivity or prognostic validity has not been evaluated. Fourth, a methodological limitation concerns the convergent validity analysis, which combined ARAT scores from two separate study groups with different measurement protocols. The correlation analysis included first-trial measurements from the reliability testing group (n = 33) and single measurements from the validity testing group (n = 31). Although both groups consisted of stroke patients meeting identical inclusion criteria, this approach may have introduced subtle differences in measurement conditions or unmeasured confounding variables between groups. This design choice, although maximizing statistical power, represents a departure from optimal validation methodology, where convergent validity should ideally be assessed within a single, homogeneously measured cohort.

## 5. Conclusions

The findings provide evidence-based support for the clinical implementation of the new ARAT as a replacement for the conventional version. The equivalent measurement properties ensure continuity in clinical practice and research, maintaining the established psychometric foundation that has made ARAT a gold-standard assessment tool internationally.

## Supporting information

**S1 Protocol. Study protocol (English version).**
(PDF)

**S2 Protocol. Study protocol (Japanese version).**
(PDF)

**S3 Checklist. CONSORT 2025 checklist.**
(PDF)

**S4 Appendix. Photographs of the new and conventional ARAT sets.** (a) New ARAT equipment, (b) Conventional ARAT equipment, (c) New ARAT Platform, (d) Conventional ARAT Platform. ARAT, Action Research Arm Test.
(PDF)

**S5 Table. Specifications of the main equipment.**
(DOCX)

**S6 Table. Specifications of the auxiliary equipment.**
(DOCX)

**S7 Table. Correlations between the Action Research Arm Test total score and clinical measures in the reliability and validity testing groups.**
(DOCX)

## Acknowledgments

We would like to thank all occupational therapists of the Department of Rehabilitation, Jikei University School of Medicine, for their cooperation in obtaining operational approval and data for conducting this study.

## Author contributions

**Conceptualization:** Daigo Sakamoto, Toyohiro Hamaguchi.

**Data curation:** Daigo Sakamoto, Toyohiro Hamaguchi.

**Formal analysis:** Daigo Sakamoto, Toyohiro Hamaguchi.

**Funding acquisition:** Daigo Sakamoto.

**Investigation:** Daigo Sakamoto.

**Methodology:** Daigo Sakamoto, Toyohiro Hamaguchi, Yasuhide Nakayama, Masahiro Abo.

**Project administration:** Masahiro Abo.

**Resources:** Yasuhide Nakayama, Masahiro Abo.

**Software:** Yasuhide Nakayama.

**Supervision:** Yasuhide Nakayama, Masahiro Abo.

**Validation:** Yasuhide Nakayama.

**Visualization:** Toyohiro Hamaguchi.

**Writing – original draft:** Daigo Sakamoto.

**Writing – review & editing:** Toyohiro Hamaguchi, Yasuhide Nakayama, Masahiro Abo.

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
