## [Decision Letter · Decision Letter 0]

15 Dec 2025

Dear Dr. Hamaguchi,

We look forward to receiving your revised manuscript.

Kind regards,

Monika Błaszczyszyn

Academic Editor

PLOS One

2. We note that you have selected “Clinical Trial” as your article type. PLOS ONE requires that all clinical trials are registered in an appropriate registry (the WHO list of approved registries is at

https://www.who.int/clinical-trials-registry-platform/network/primary-registries " https://www.who.int/clinical-trials-registry-platform/network/primary-registries

and more information on trial registration is at http://www.icmje.org/about-icmje/faqs/clinical-trials-registration/ ). Please state the name of the registry and the registration number (e.g. ISRCTN or

ClinicalTrials.gov ) in the submission data and on the title page of your manuscript. a) Please provide the complete date range for participant recruitment and follow-up in the methods section of your manuscript. b) If you have not yet registered your trial in an appropriate registry, we now require

you to do so and will need confirmation of the trial registry number before we can pass your paper to the next stage of review. Please include in the Methods section of your paper your reasons for not registering this study before enrolment of participants started. Please confirm that all related trials are

registered by stating: “The authors confirm that all ongoing and related trials for this drug/intervention are registered”. Please see http://journals.plos.org/plosone/s/submission-guidelines#loc-clinical-trials for our policies on clinical trials.

[JSPS KAKENHI (Grant number: JP 24K14384)].

Additional Editor Comments (if provided):

Reviewers' comments:

Reviewer's Responses to Questions

**Comments to the Author**

1. Is the manuscript technically sound, and do the data support the conclusions?

Reviewer #1: Yes

Reviewer #2: Yes

2. Has the statistical analysis been performed appropriately and rigorously?

Reviewer #1: Yes

Reviewer #2: Yes

3. Have the authors made all data underlying the findings in their manuscript fully available?

Reviewer #1: No

Reviewer #2: No

4. Is the manuscript presented in an intelligible fashion and written in standard English?

Reviewer #1: Yes

Reviewer #2: Yes

Reviewer #1: The research team enrolled 64 patients with stroke and hemiparesis to evaluate the reliability and validity of the newly developed ARAT. The results showed the new ARAT has excellent reliability and strong criterion-related validity compared with the conventional version.

1. While the new ARAT maintained identical assessment protocols, modifications in several test objects existed. Please explain the reason for incorporating these modifications and comment on their potential effects on performance and interpretation of final results.

2. It would be good to include p-values for formal group comparison in Table 1.

3. Convergent validity was assessed based on two groups, which may introduce confounding. Is there any way to evaluate its impact or can any potential sensitivity analysis be conducted?

4. Figures 4 and 5 show some data points aligning to either the vertical bar or the horizontal bar. Is there a potential threshold effect? How does it affect the results and interpretation?

Reviewer #2: I laud the authors on their well-crafted manuscript of an expertly designed and responsibly executed study!

Please see the attached .pdf file for my detailed commentary (which I hope that the authors will be capable of addressing).

**Do you want your identity to be public for this peer review?** For information about this choice, including consent withdrawal, please see our Privacy Policy

Reviewer #1: No

Reviewer #2: **Yes:** 0ge arum

---

## [Author Response · Author response to Decision Letter 1]

26 Jan 2026

We wrote our responses to the reviwers in response letters and submitted files: 20260122_Response_reviewer_1.docx and 20260122_Response_reviewer_2.docx.

Please fined these files.

---

## [Decision Letter · Decision Letter 1]

24 Feb 2026

Reliability and validity of a newly developed Action Research Arm Test for upper limb function assessment in patients with stroke: A comparison with the conventional version

PONE-D-25-51652R1

Dear Dr. Hamaguchi,

We’re pleased to inform you that your manuscript has been judged scientifically suitable for publication and will be formally accepted for publication once it meets all outstanding technical requirements.

Kind regards,

Monika Błaszczyszyn

Academic Editor

PLOS One

Reviewers' comments:

Reviewer's Responses to Questions

**Comments to the Author**

Reviewer #1: All comments have been addressed

Reviewer #2: All comments have been addressed

2. Is the manuscript technically sound, and do the data support the conclusions?

Reviewer #1: (No Response)

Reviewer #2: Yes

3. Has the statistical analysis been performed appropriately and rigorously?

Reviewer #1: (No Response)

Reviewer #2: Yes

4. Have the authors made all data underlying the findings in their manuscript fully available?

Reviewer #1: (No Response)

Reviewer #2: Yes

5. Is the manuscript presented in an intelligible fashion and written in standard English?

Reviewer #1: (No Response)

Reviewer #2: Yes

Reviewer #1: (No Response)

Reviewer #2: The authors conscientiously addressed all of my comments, some of which were (reassuringly) convergent with those of the other reviewer; it was an honor to peer-review their well-written manuscript on their well-executed study!

**Do you want your identity to be public for this peer review?** For information about this choice, including consent withdrawal, please see our Privacy Policy

Reviewer #1: No

Reviewer #2: **Yes:** 0ge arum, Ph.D.

---

## [Editor Report · Acceptance letter]

PONE-D-25-51652R1

PLOS One

Dear Dr. Hamaguchi,

I'm pleased to inform you that your manuscript has been deemed suitable for publication in PLOS One. Congratulations! Your manuscript is now being handed over to our production team.

Kind regards,

on behalf of

Dr. Monika Błaszczyszyn

Academic Editor

PLOS One